# The SARS-CoV-2 Variant Omicron Is Able to Escape Vaccine-Induced Humoral Immune Responses, but Is Counteracted by Booster Vaccination

**DOI:** 10.3390/vaccines10050794

**Published:** 2022-05-17

**Authors:** Florian D. Hastert, Sascha Hein, Christine von Rhein, Nuka Ivalu Benz, Younes Husria, Doris Oberle, Thorsten J. Maier, Eberhard Hildt, Barbara S. Schnierle

**Affiliations:** 1Department of Virology, Section AIDS and Newly Emerging Pathogens, Paul-Ehrlich-Institut, 63225 Langen, Germany; Florian.Hastert@pei.de (F.D.H.); Christine.vonRhein@pei.de (C.v.R.); 2Department of Virology, Paul-Ehrlich-Institut, 63225 Langen, Germany; Sascha.Hein@pei.de (S.H.); NukaIvalu.Benz@pei.de (N.I.B.); Younes.Husria@pei.de (Y.H.); Eberhard.Hildt@pei.de (E.H.); 3Division of Pharmacovigilance, Paul-Ehrlich-Institut, 63225 Langen, Germany; Doris.Oberle@pei.de (D.O.); ThorstenJuergen.Maier@pei.de (T.J.M.)

**Keywords:** SARS-CoV-2, Delta, Omicron, vaccine, neutralization

## Abstract

The SARS-CoV-2 variant Omicron has spread world-wide and is responsible for rapid increases in infections, including in populations with high vaccination rates. Here, we analysed in the sera of vaccinated individuals the antibody binding to the receptor-binding domain (RBD) of the spike protein and the neutralization of wild-type (WT), Delta (B.1.617.2), and Omicron (B.1.1.529; BA.1) pseudotyped vectors. Although sera from individuals immunized with vector vaccines (Vaxzevria; AZ and COVID-19 Janssen, Ad26.COV2.S; J&J) were able to bind and neutralize WT and Delta, they showed only background levels towards Omicron. In contrast, mRNA (Comirnaty; BNT) or heterologous (AZ/BNT) vaccines induced weak, but detectable responses against Omicron. While RBD-binding antibody levels decreased significantly six months after full vaccination, the SARS-CoV-2 RBD-directed avidity remained constant. However, this still coincided with a significant decrease in neutralization activity against all variants. A third booster vaccination with BNT significantly increased the humoral immune responses against all tested variants, including Omicron. In conclusion, only vaccination schedules that included at least one dose of mRNA vaccine and especially an mRNA booster vaccination induced sufficient antibody levels with neutralization capacity against multiple variants, including Omicron.

## 1. Introduction

The severe acute respiratory syndrome coronavirus 2 (SARS-CoV-2) pandemic has resulted in the development of five virus variants of concern (VOC), Alpha, Beta, Gamma, Delta, and Omicron. The selection of these variants was mainly towards viruses that are more transmissible and, therefore, more contagious, such as Alpha and Delta. While partial immune escape from convalescent or vaccinated individuals has also been described for the Beta, Gamma, and Delta variants [1,2,3], this changed dramatically with the appearance of the Omicron variant. Omicron became the globally dominant variant within two months of its first description in late 2021 and replaced the previously dominant Delta variant. For the first time in the SARS-CoV-2 pandemic, high numbers of reinfections of convalescent and vaccinated individuals were observed [4]. Omicron has a large number of amino acid substitutions, insertions, and deletions in the viral spike protein compared to the original Wuhan-Hu-1 virus (WT), of which alone 15 are found within its receptor-binding domain (RBD). This abundance of mutations apparently reflects the evolution of a new SARS-CoV-2 serotype [5].

Polyclonal neutralizing antibody responses target the RBD and the N-terminal (NTD) domains of the viral spike protein [6,7]. Their levels are predictive of immune protection from symptomatic SARS-CoV-2 infection [8]. However, Omicron has 35 mutations in its spike protein, suggesting a massive immune escape. The Omicron spike shows increased binding to the ACE2 receptor, which has been studied on a protein structural basis [9,10]. Since the spike protein is the central antigen used to induce an immune response in all authorized vaccines, we studied the recognition of WT, Delta, and Omicron SARS-CoV-2 by vaccine-elicited antibodies. Furthermore, we compared differences in vaccination schedules and vaccines used. To date, between 50 and 70% booster vaccinations have been administered in countries with unrestricted access to COVID-19 vaccines since the beginning of 2021, such as, for example, Germany or the U.K. The data of the respective health authorities leave between a third up to half of the respective populations more vulnerable to SARS-CoV-2 infections by Omicron-derived variants due to the lack of a booster vaccination. The present study may therefore help to estimate the risk of infection in the context of different vaccination schedules.

## 2. Materials and Methods

### 2.1. Cell Culture

HEK293T (ATCC CRL-3216) and HEK293T-hACE2 cells [11] were grown in Dulbecco’s Modified Eagle Medium (DMEM; Sigma, Taufkirchen, Germany) supplemented with 10% foetal bovine serum (Sigma, Taufkirchen, Germany), 5% l-glutamine (200 mM; Lonza, Verviers, Belgium), and 1% penicillin/streptavidin (Fisher Scientific, Schwerte, Germany) at 37 °C under 5% CO_2_. The medium for HEK293T-hACE2 cells was furthermore supplemented with 50 µg/mL Zeocin™ (Fisher Scientific, Schwerte, Germany).

### 2.2. Serum Samples

Vaccinated and previously SARS-CoV-2 naïve individuals, with no relevant underlying health predispositions, comprised 9 female donors between 28 and 63 years and 14 male donors between 27 and 66 years of age. The first blood sampling took place directly before the second dose for the AZ/AZ, AZ/BNT, and BNT/BNT vaccination schedules and before the single dose of the J&J vaccination (Day 0). The first dose of AZ was given 56 days before the second dose, and double-BNT-vaccinated individuals received their first dose 21 days before their second vaccination. Follow-up samples were collected two weeks later (Day 14) and six months after the second dose, right before the third dose for the AZ/BNT/BNT and BNT/BNT/BNT vaccinations (Day 180). From these individuals, an additional sample was collected 14 days after the booster dose (Day 194). Figure 1 summarizes the blood collections and vaccinations.

### 2.3. Ethics and Study Design

The retro-prospective PEI-SARS-CoV-2 study was reviewed and approved by the Ethics Committee of the Landesärztekammer Hessen, Germany (Ethikvotum 2020-1664-3-evBO). The study was performed at the Paul-Ehrlich-Institut, Langen, Germany. Subjects were recruited either at a vaccination centre or at the Paul-Ehrlich-Institute prior to vaccination or second vaccination. To be included in the study, subjects had to be at least 18 years of age, had to give written informed consent, and had to be not previously been infected with COVID-19. All serum samples were screened for the presence of anti-N-antibodies and were negative.

### 2.4. ELISA

To assess the specific binding of IgG to the RBD of either WT, Delta, or Omicron SARS-CoV-2, a previously described in-house ELISA was used [12,13,14]. In brief, the SARS-CoV-2 spike RBD was transiently expressed in HEK293T cells and purified by Ni-affinity chromatography [12]. The RBD was used to coat 96-well microtiter ELISA plates (Costar 3590, Corning Incorporated, Kennebunk, NY, USA) at a concentration of 2 μg/mL overnight at 4 °C. Afterwards, the plates were blocked for 1 h at RT with 3% BSA in PBS containing 0.1% Tween-20 (PBS-T). Between each step, the plates were washed three times with PBS-T. To prevent the overloading of the ELISA binding capacity and to stay within the functional range of the assay, sera were pre-diluted 1:50 in PBS supplemented with 1% BSA and incubated for 2 h at RT. To control the amount of coated RBD on the surface, a chimeric human anti-His antibody (Clone RMH01; Biozol; Eching; Germany) was used in a dilution of 1:3000 in duplicate for each plate. For detection, HRP-linked anti-human IgG antibody (Cytiva, Dassel, Germany) was used in a 1:3000 dilution. Between each step, the plates were washed three times with PBS-T. The ELISA plates were developed with 100 μL TMB ELISA Substrate Solution (eBioscience, San Diego, CA, USA) for 5 min, stopped with 100 μL 1 N sulfuric acid, and analysed directly by measuring the absorption at 450 nm on an Infinite M1000 reader (Tecan Group, Männedorf, Switzerland). The cut-off level was determined from the mean OD_450_ values obtained with samples taken prior to the J&J vaccination, which were from naïve individuals, plus two standard deviations, and was an OD value of 0.033. All values were normalized to the values obtained with the anti-His directed antibody.

### 2.5. Avidity Measurements of IgG Using ELISA

The SARS-CoV-2-specific antibody avidity was analysed with the above-mentioned, in-house RBD ELISA. Microtiter plates were incubated with patient serum for 2 h and subsequently washed using a washing buffer with or without 4 M urea for 5 min at 37 °C. The avidity was calculated as the ratio between the optical densities obtained with and without urea wash, multiplied by 100, and was plotted as a percentage (%) [15].

### 2.6. Serum Neutralization Assay Using Pseudotyped Lentiviral Vectors

Lentiviral vectors were produced in HEK293T cells by co-transfection using Lipofectamine^®^ 2000 (Thermo Fisher, Darmstadt, Germany) as described previously [16]. Plasmids encoding HIV-1 gag/pol, rev, the luciferase-encoding lentiviral vector genome, and the SARS-CoV-2 wild-type 614D (#MN908947), the SARS-CoV-2 Delta (NC_045512.2), or the SARS-CoV-2 Omicron BA.1 (NC_045512.2) spike gene were transfected. The coronavirus spike genes were truncated, lacking the last 19 carboxy terminal amino acids, and were synthesized (Eurofins, Ebersberg, Germany; IDT, Leuven, Belgium) and cloned into the vector pcDNA 3.1^(+)^ as described before [17]. After harvest, vector particles were used directly for neutralization assays. Pseudotyped vectors and serially diluted human serum (1:60 to 1:4860) were incubated for 45 min at 37 °C and used to transduce HEK293T-hACE2 cells in triplicate [12]. After 48 h, britelite plus luciferase substrate (PerkinElmer, Waltham, MA, USA) was added to measure luciferase activity. The reciprocal area under the curve (1/AUC) value calculated for each sample corresponds to the respective neutralization activity. Cut-off levels were determined with samples taken prior to the J&J vaccination, constituted by their mean values plus two standard deviations, and had a 1/AUC of 0.45 for WT, 0.44 for Delta, and 0.20 for Omicron.

### 2.7. Statistical Analysis and Software

Area under the curve (AUC) values were determined using the GraphPad Prism 7.04 software (La Jolla, CA, USA). In brief, mean values from triplicate experiments and the corresponding standard deviations were depicted in the graphs. For this, 0 was chosen as the baseline, and only peaks above the baseline were taken into account. Neutralization and antibody levels were analysed and plotted with RStudio Version 2022.02.0 Build 443 (https://www.R-project.org/ accessed on 17 March 2022). Statistical significance was determined with RStudio Version 2022.02.0 Build 443 using a Wilcoxon rank sum test. Significance is indicated with two stars (**) for a *p*-value between 0.001 and 0.01, and one star (*) denotes a *p*-value between 0.01 and 0.05. Fold changes were calculated by subtracting 1 from the quotient of the final value Y and the initial value X: ((Y/X) − 1).

## 3. Results

### 3.1. Serum Samples

Serum samples were collected from 23, SARS-CoV-2 vaccinated persons who were confirmed to be SARS-CoV-2 naïve by N-ELISA of all samples tested. One cohort of 11 individuals was vaccinated with adenovirus-based vaccines, Vaxzevria (AZ), or Ad26.COV2.S (J&J). Six people received two doses of Vaxzevria (AZ/AZ) separated by 56 days. Blood was drawn directly before and 14 days after the second dose. Another five individuals received one dose of Ad26.COV2.S (J&J). Samples were also taken from this group before their single-dose vaccination and served as negative controls and determined the background values of the assays (Figure 1).

Another 12 individuals were vaccinated with the mRNA vaccine Comirnaty (BNT) or a combination of AZ and BNT. Six people received a first dose of AZ and two additional doses of BNT (AZ/BNT/BNT). The BNT dose was given 56 days after the AZ vaccination, and the third dose was given six months after the second. Another six people received three doses of Comirnaty (BNT/BNT/BNT). The first two doses were separated by 21 days, and the third dose was again given six months after the second. Blood was drawn directly before and 14 days after the second dose. Furthermore, blood samples were collected six months after the second dose, right before the third dose and, finally, 14 days later (Figure 1). The cohort comprised nine female participants aged 28 to 63 and 14 male individuals aged 27 to 66 (Table 1).

### 3.2. Humoral Immune Response against SARS-CoV-2 Variants of Individuals Vaccinated with Vector Vaccines

First, the serum samples of individuals vaccinated with either of the two vector vaccines, AZ and J&J, were analysed for the presence of IgG binding to the SARS-CoV-2 spike receptor-binding domain (RBD). To this end, a previously described in-house ELISA, coated with the RBD from WT, Delta, or Omicron SARS-CoV-2, was established and used [12].

The first dose of AZ induced WT and Delta RBD binding IgG with mean OD values of 0.26 ± 0.13 and 0.21 ± 0.10, respectively. However, a second vaccination with AZ significantly increased the levels of WT RBD-binding IgGs compared to the first dose of AZ (*p* < 0.05) (Figure 2A). Comparable results were found for the Delta RBD (Figure 2A), with mean OD_450_ values of 0.46 ± 0.16 for WT and 0.41 ± 0.15 for Delta. The levels of WT and Delta RBD-binding antibodies were significantly higher (*p* < 0.01) after two doses of AZ compared to one dose of J&J with mean OD_450_ values 0.46 ± 0.16 versus 0.30 ± 0.12 for WT and 0.41 ± 0.15 versus 0.23 ± 0.15 for Delta (Figure 2A, J&J in yellow). One-dose AZ induced similar antibody levels as a J&J vaccination. Neither vaccination with a vector vaccine induced Omicron RBD-binding antibodies above background levels (Figure 2A).

Furthermore, the IgG-binding avidity was determined by 4 M urea washes in the binding ELISA assays. Due to the low levels of Omicron-RBD-binding antibodies, no avidity could be calculated for this variant. For WT and Delta, however, respective antibody avidities were measured (Figure 2B). Two doses of AZ yielded the highest avidity with a value of 98.80 ± 15.28%, which was significantly higher than the values obtained after one dose of AZ (76.23 ± 5.22%; *p* < 0.005) or one dose of J&J (39.75 ± 6.65%; *p* < 0.005) (Figure 2B).

The results from the analysis of binding antibody levels at Day 14 after the second vaccination were largely reaffirmed in the neutralization capacity of the respective sera. This was analysed with pseudotyped lentiviral vectors as described before, using the WT, Delta, or Omicron spike proteins [12]. A single dose of AZ was only sufficient to induce neutralizing antibodies above the background towards WT pseudotyped vectroparticles (mean 1/AUC: 0.59 ± 0.11), but not against Delta or Omicron. Two doses of AZ resulted in neutralization potential towards WT- and Delta-pseudotyped vectors with mean reciprocal area under the curve (1/AUC) values of 0.97 ± 0.30 and 0.68 ± 0.33, respectively (Figure 3). In contrast, neutralization of Omicron pseudotyped vector particles was with a value of 0.24 ± 0.09 very close to the background level of 0.20, and therefore not evident. One dose of the J&J vaccine resulted in WT neutralizing potential that was significantly increased compared to the sera of persons with two doses of AZ (*p* < 0.01; mean = 2.02 ± 0.59), but neutralization was comparably low towards Delta (mean 0.68 ± 0.12) (Figure 3). Corresponding to RBD-binding antibody levels, no neutralization of Omicron above background levels was detected (mean value 0.23 ± 0.02) (Figure 3).

In summary, vector vaccines induced WT- and Delta-RBD-binding IgG at 14 days after the final vaccination with increasing antibody avidity over time. A single dose of J&J vaccine was superior to two doses of AZ in the neutralization of WT, but not Delta. Omicron pseudotyped vector particles were not neutralized by any serum, and antibodies binding the Omicron RBD were also not detected.

### 3.3. Humoral Immune Response against SARS-CoV-2 Variants of Persons Vaccinated with mRNA or Heterologous Vaccination Schedules

The mRNA vaccination was tested in two settings either as BNT alone or as a heterologous vaccination of AZ priming and BNT boosts. The vaccinees were followed for six months and received a third booster dose. Blood was sampled immediately before the second dose, 14 days after the second dose, 6 months after the second dose, on the day of the booster dose, and, finally two weeks after the booster dose (Figure 1).

As described above, a single dose of AZ induced low levels of WT RBD-binding IgG (mean OD_450_ = 0.24 ± 0.09). After receiving BNT as a second dose, WT RBD-directed antibody levels reached a mean OD_450_ of 0.88 ± 0.04, which was similar to that reached after two doses of BNT (mean OD_450_ = 0.89 ± 0.10; Figure 4). Moreover, a first dose of BNT elicited by far the highest initial RBD-binding antibody levels (mean OD_450_ = 0.55 ± 0.24). These levels were significantly higher than those obtained after a first dose of AZ (mean OD_450_ = 0.24 ± 0.09; *p* < 0.05), but not after a single dose of J&J (mean OD_450_ = 0.30 ± 0.12). For the Delta RBD, however, a first dose of BNT (mean OD_450_ = 0.55 ± 0.27) elicited levels of RBD-binding antibodies about two-times higher than a first dose of AZ (mean OD_450_ = 0.24 ± 0.10; *p* < 0.01) (Figure 4). Interestingly, Omicron RBD-specific IgG levels, although very low, were significantly higher (*p* < 0.005) in heterologously vaccinated individuals compared to vaccinees, who had received two doses of BNT (mean OD_450_ 0.12 ± 0.06 compared to 0.03 ± 0.009 respectively) (Figure 4).

Overall, the RBD-binding antibody levels waned over time at similar rates, while the differences in variant-specific antibody responses were retained. After six months, the values for WT (mean OD_450_ = 0.57 ± 0.19) and Delta RBD binding (mean OD_450_ = 0.53 ± 0.21) for individuals vaccinated twice with BNT decreased to levels similar to the first sampling time point (Figure 4). In contrast, heterologously vaccinated individuals retained a decreased, but still significantly higher binding against WT (mean OD_450_ = 0.54 ± 0.15; *p* < 0.01) and Delta (mean OD_450_ = 0.58 ± 0.16; *p* < 0.005) compared to the initial AZ vaccination (Figure 4). Since Omicron RBD-binding antibodies were close to background levels, no significant difference was detected after 180 days.

The booster vaccination after six month significantly increased the levels of RBD-binding antibodies against all three variants and reached levels that were slightly higher than those reached after the second vaccination for WT- and Delta-RBD in both vaccination schedules. Interestingly, heterologous and also homologous vaccinations increased the levels of Omicron-RBD-binding IgG to similarly low, but detectable levels (mean OD_450_ = 0.18 ± 0.10 and 0.19 ± 0.18, respectively). The overall values after the boost were the same for both vaccination schedules.

Antibody avidity against WT- and Delta-RBD was analysed as before. Avidity for WT RBD (mean = 87.3 ± 8.44% to 110.3 ± 3.28% for AZ/BNT/BNT; *p* < 0.005 and 80.7 ± 10.42% to 103.5 ± 5.52% for BNT/BNT/BNT; *p* < 0.005) and Delta (mean = 71.6 ± 7.75% to 103.2 ± 5.10% for AZ/BNT/BNT; *p* < 0.005 and 70.6 ± 8.49% to 106.1 ± 15.21% for BNT/BNT/BNT; *p* < 0.005) significantly increased at Day 14 and plateaued thereafter. Antibody avidity of Omicron-RBD binding could not be calculated, due to a lack of detectable binding. The booster vaccination did not significantly change antibody avidities (Figure 5).

A similar pattern to that observed for antibody binding was also found for neutralizing antibodies. The second vaccination increased the amount of neutralizing antibodies directed against WT- and Delta-pseudotyped vectors, with BNT/BNT showing two- to three-fold higher neutralization (mean 1/AUC = 7.71 ± 4.72 for WT and 2.51 ± 2.17 for Delta) (Figure 6) at 14 days after the second vaccination than AZ/BNT-immunized individuals (mean 1/AUC = 3.33 ± 2.17 for WT and 1.50 ± 0.63 for Delta). At six months, the neutralization activity against WT and Delta decreased to very low levels (mean WT: AZ/BNT 1.22 ± 0.76 and BNT/BNT 0.80 ± 0.32). However, it was increased by the booster vaccination to similar or slightly higher levels compared to 14 days after the second vaccination. The heterologous vaccination always produced lower values than the mRNA vaccinations (mean 3.70 ± 1.18 versus 4.35 ± 1.37, respectively). Neutralization of Omicron was virtually absent in all samples, except for those taken after the booster vaccination. No difference in Omicron-specific neutralization activity could be detected between the vaccinations in the final booster values (mean 1.10 ± 0.81 for AZ/BNT/BNT and 1.12 ± 0.70 for BNT/BNT/BNT).

In conclusion, although antibody avidity was constantly high at 14 days after the second vaccination, a significant drop in RBD-binding and neutralization activity was observed after six months. The third vaccination increased RBD-binding and neutralization activity and, for the first time, resulted in the recognition of the Omicron-RBD and Omicron-neutralization activity.

## 4. Discussion

The first COVID-19 vaccines that received a marketing authorization were mRNA vaccines developed by Pfizer/BioNTech and Moderna. Subsequently, two adenoviral vector vaccines from AstraZeneca (AZ) and Johnson & Johnson (J&J) were approved. All four COVID-19 vaccines showed high efficacy in clinical trials and real-world data. Vaccine efficacy was shown to correlate with neutralizing antibody titres in vitro, and with 90–95% efficacy, mRNA vaccines are more efficacious than adenovirus-based vaccines with 75–80% [18,19]. Our data confirm the differences between the various vaccines in healthy adults aged 27–66. At 14 days after full vaccination, the mRNA vaccination resulted in the highest binding and neutralization activity against the WT and the Delta SARS-CoV-2 variants, followed by the heterologous vaccination, while the double AZ vaccinations and the single dose of J&J vaccine gave rise to the lowest values.

Vaccine efficacy data were obtained with WT SARS-CoV-2, and efficacy decreased with the appearance of variants. We therefore tested the sera of differently vaccinated individuals for the ability of the induced antibodies to bind and neutralize either WT or the Delta or Omicron variants. As others before, we saw only a minimal decrease in RBD binding (1.4-fold ±0.29) and neutralization (2.6-fold ±1.1) of the Delta variant [20]. The mRNA vaccine and the heterologous vaccination schedule were superior with a two-fold higher neutralization activity against WT and Delta compared to the adenovirus-based vaccines. We cannot totally exclude that these differences are due to sex differences in the different vaccine groups, because the adenovirus-based vaccine group comprised mainly male participants and the mRNA group mainly females. Interestingly, male convalescent COVID-19 patients had greater antibody titres than females and real-world vaccine efficacy seems to be greater in men [21,22]. However, all responses waned over time and were barely detectable after six months. In contrast, antibody avidity increased after the second vaccination and stayed at this level during the six-month follow-up observation time.

Strikingly, Omicron-RBD binding and neutralization were severely impaired and at or close to background levels. A third vaccination, however, significantly increased the levels of neutralizing and binding antibodies against all three SARS-CoV-2 variants. In particular, only these boosted individuals had binding and neutralizing antibodies against Omicron, although at low levels (mean reciprocal AUC = 1.11 ± 0.72; mean OD_450_ = 0.19 ± 0.14). Considering the observed high infection rates of convalescent and vaccinated people for Omicron, our data support previous findings that neutralizing antibody levels are predictive of the immune protection from SARS-CoV-2 infections [8]. A limitation of the study is the small number of participants and potential effects from sex differences that cannot be accounted for in this study cohort. In addition, participants were healthy and between 27 and 65 year of age, which does not allow for conclusions on immune responses of individuals with comorbidities or children.

Our data confirm several recent publications describing a strong immune evasion of Omicron from vaccination and SARS-CoV-2-specific monoclonal antibodies, as well as in recovered patients [23,24,25,26,27,28,29]. Although Omicron is hardly neutralized in vitro, most reinfected individuals appear to have only mild disease symptoms, indicating that other mechanisms contribute to protection from severe disease, such as cellular immune responses, which have been described to be highly conserved [30,31,32,33], or Fc-receptor-dependent responses, which are also conserved against Omicron [34].

## 5. Conclusions

The analysis of humoral immune responses presented here suggests that the emergence of Omicron is due to immune selective pressure in the hosts [28]. Although a booster vaccination results in an increased recognition of Omicron at low levels, broadly protective pan-sarbecovirus vaccines are urgently needed.

## Figures and Tables

**Figure 1 vaccines-10-00794-f001:**
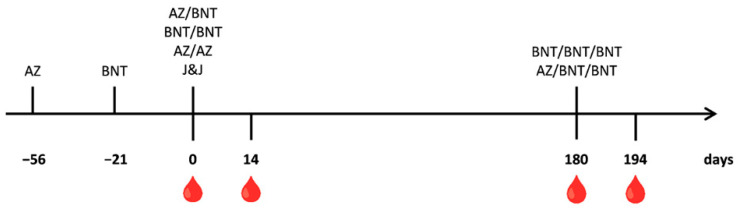
**Schematic presentation of the SARS-CoV-2 vaccinations and blood sampling time points.** Blood sampling is indicated by a blood drop. Vaccinations are shown as letters. The time point 0 of the J&J vaccination corresponds to naïve sera and was used to determine the background levels of the assays.

**Figure 2 vaccines-10-00794-f002:**
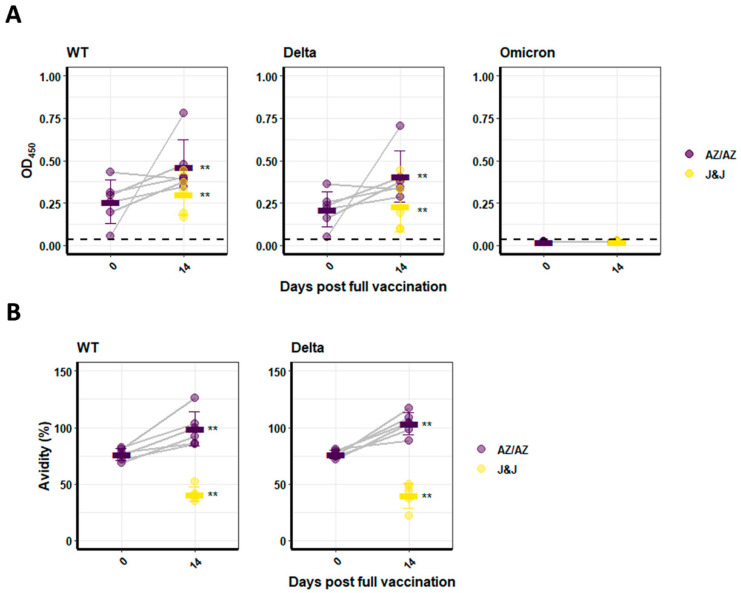
**Analysis of RBD binding and IgG avidity in sera**. (**A**) IgG binding to the WT, Delta, or Omicron RBD was analysed by ELISA. Binding is depicted as OD_450_ values. Binding of IgG after the first and the second AZ vaccination is shown. J&J was applied as a single dose and is depicted in yellow. (**B**) IgG avidity measurements after one or two doses of AZ or one dose of J&J compared to preimmune serum. Omicron RBD avidity could not be determined as it was below background values. Bars indicate the arithmetic mean for each group. Significance is indicated with two stars (**) for a *p*-value between 0.001 and 0.01.

**Figure 3 vaccines-10-00794-f003:**
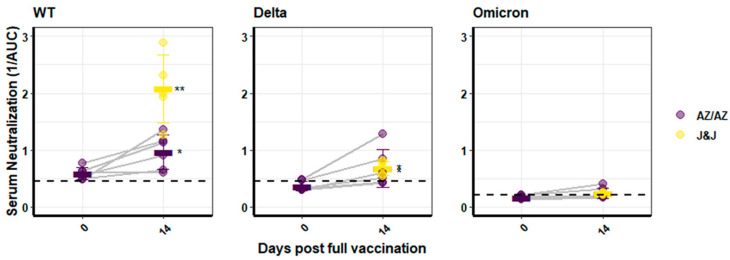
**Neutralization activity of sera.** Sera from AZ- or J&J-immunized individuals were analysed using lentiviral vectors pseudotyped with the WT, Delta, or Omicron spike. Neutralization is depicted as the reciprocal area under the curve. J&J vaccination was a single dose, and the values are depicted in yellow. For Omicron, no significance was assessed, as mean levels were below background values. Bars indicate the arithmetic mean for each group. Significance is indicated with two stars (**) for a *p*-value between 0.001 and 0.01, and one star (*) denotes a *p*-value between 0.01 and 0.05.

**Figure 4 vaccines-10-00794-f004:**
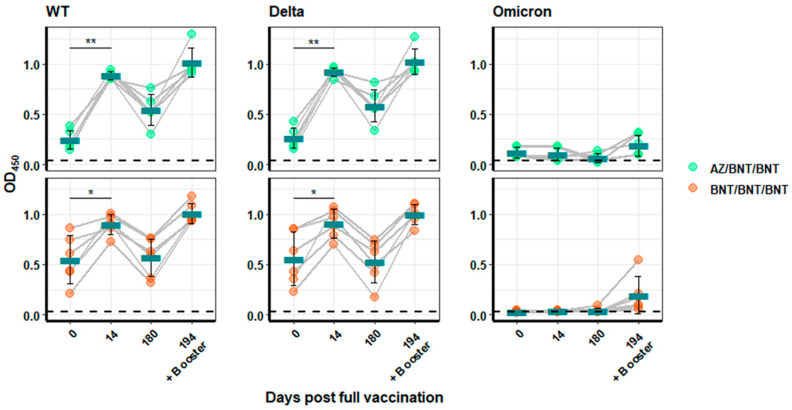
**Analysis of RBD binding in sera of heterologous or BNT-vaccinated individuals**. IgG binding to the WT-, Delta-, or Omicron-RBD in sera from individuals who received either AZ/BNT/BNT (green) or BNT/BNT/BNT (orange) was analysed by ELISA. Blood samples were each collected at Days 0, 14, and 180 post full immunization and, furthermore, 14 days after a third booster dose. Binding is depicted as OD_450_ values. Turquoise bars indicate the arithmetic mean for each group. Significance is indicated with two stars (**) for a *p*-value between 0.001 and 0.01, and one star (*) denotes a *p*-value between 0.01 and 0.05.

**Figure 5 vaccines-10-00794-f005:**
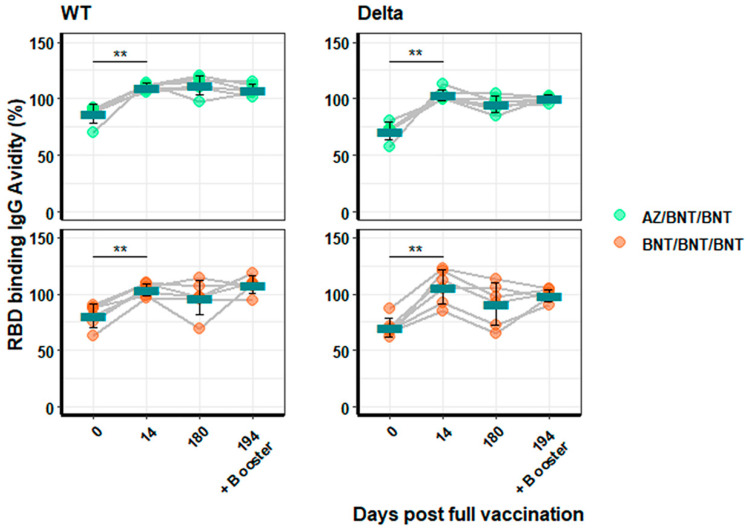
**Analyses of IgG avidity to RBD in sera from heterologous or BNT-vaccinated individuals**. Depicted are IgG avidity measurements for WT- and Delta-RBD binding from individuals vaccinated AZ/BNT/BNT (green) and BNT/BNT/BNT (orange) immunization schedules. Blood was collected at 0, 14, and 180 days post full vaccination and, additionally, 14 days after the booster immunization. Omicron-RBD avidity could not be determined, because it was below background values. Turquoise bars indicate the arithmetic mean for each group. Significance is indicated with two stars (**) for a *p*-value between 0.001 and 0.01.

**Figure 6 vaccines-10-00794-f006:**
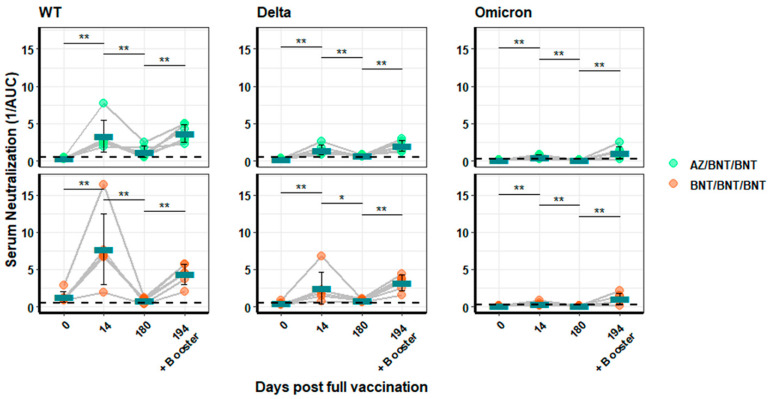
**Neutralization activity of sera from heterologous or BNT-immunized individuals.** Sera from AZ/BNT/BNT (green) or BNT/BNT/BNT (orange) immunized individuals were analysed using lentiviral vectors pseudotyped with the WT, Delta, or Omicron spike protein. Neutralization is depicted as the reciprocal area under the curve. Blood was drawn at Days 0, 14, and 180 after full vaccination and, in addition, at Day 194, 14 days post booster vaccination. Turquoise bars indicate the arithmetic mean for each group. Significance is indicated with two stars (**) for a *p*-value between 0.001 and 0.01, and one star (*) denotes a *p*-value between 0.01 and 0.05.

**Table 1 vaccines-10-00794-t001:** Demographics of vaccinated individuals.

Vaccination Schedule	Age	Sex
AZ/AZ	27–33	0 female, 5 male
J&J	31–64	1 female, 5 male
AZ/BNT/BNT	41–63	3 female, 3 male
BNT/BNT/BNT	28–60	5 female, 1 male

All tested individuals are of Caucasian descent.

## Data Availability

The data presented in this study are available upon request from the corresponding author. The data are not publicly available due to ethical restrictions.

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
