# Peer review of "The SARS-CoV-2 Variant Omicron Is Able to Escape Vaccine-Induced Humoral Immune Responses, but Is Counteracted by Booster Vaccination"

_vaccines, 2022, doi:10.3390/vaccines10050794_

Round 1

Reviewer 1 Report

Comments for Vaccines-1696105

The manuscript is not appropriate at this stage to be published in Vaccines unless the following points are addressed.

  1. Modify the title as the study focus on both escaping powers of Omicron of vaccine-induced immune and efficacy of Booster.
  2. In the abstract, the authors mentioned: “only a booster vaccination induces sufficient antibody levels with neutralization capacity against multiple variants, including Omicron”. Then what is the necessity/contribution of other dosages that are taken before Booster?
  3. The authors should add a comparison study between the mentioned variants and Booster and other dosages. It clarifies the study more efficiently.
  4. The Introduction is too short. It’s needs a more recent literature survey in support of the study.
  5. The authors collected the samples from the people of 27/28-63/66 ages. It means this study is not valid for people of all ages, especially for young (15-25) ages. Justify it in the manuscript.
  6. In figure 5, “the booster vaccination did not significantly change antibody avidities to RBD in sera from heterologous or BNT vaccinated individuals.” So, if the booster doesn’t change antibody avidity, how could it induce sufficient antibody levels?
  7. Extensive English editing is necessary.
  8. Overall, this is an interesting study. But here I found restrictions about some facts, like
  9. Age restriction
  10. People suffering from other diseases
  11. Pregnant women

The authors should justify the study for the above fact.

Author Response

Barbara Schnierle

Department of Virology

Section AIDS and newly emerging pathogens

Telefon / Phone +49 (0) 6103 77 5504

Fax: +49 (0) 6103 77 1234

E-Mail: [email protected]

May 11th, 2022

Paul-Ehrlich-Institut  Postfach  63207 Langen

Dear Reviewer,

First, we would like to thank you and the reviewers for the excellent comments and for giving us the opportunity to revise the manuscript. Below we listed the points raised by the reviewers, our response to them and the actions we took to advance the work.

Reviewer 1

The manuscript is not appropriate at this stage to be published in Vaccines unless the following points are addressed.

  1. Modify the title as the study focus on both escaping powers of Omicron of vaccine-induced immune and efficacy of Booster.

àAs suggested by the Reviewer, the title was adapted to: The SARS-CoV-2 variant Omicron is able to escape vaccine-induced humoral immune responses but counteracted by booster vaccination.

  1. In the abstract, the authors mentioned: “only a booster vaccination induces sufficient antibody levels with neutralization capacity against multiple variants, including Omicron”. Then what is the necessity/contribution of other dosages that are taken before Booster?

à Thank you for the important comment. We changed the sentence in line 29 to: “In conclusion, only vaccination schedules that included at least one dose of mRNA vaccine and especially an mRNA booster vaccination induced sufficient antibody levels with neutralization capacity against multiple variants, including Omicron.”

  1. The authors should add a comparison study between the mentioned variants and Booster and other dosages. It clarifies the study more efficiently.

à We emphasized the differences between standard vaccinations and booster vaccination by changing the sentence starting in line 381 to:” The mRNA vaccine and the heterologous vaccination schedule were superior with a two-fold higher neutralization activity against WT and Delta compared to the adenovirus-based vaccines.”

  1. The Introduction is too short. It’s needs a more recent literature survey in support of the study.

àWe included more information in the Introduction, starting at line 48 and two more references #9 and #10. We explained in more detail the reasons to perform the study and the major alterations in the Omicron spike protein.

  1. The authors collected the samples from the people of 27/28-63/66 ages. It means this study is not valid for people of all ages, especially for young (15-25) ages. Justify it in the manuscript.

à We included this limitation of our data in the Discussions section in line 372:” Our data confirm the differences between the various vaccines in healthy adults age 27-66.”

  1. In figure 5, “the booster vaccination did not significantly change antibody avidities to RBD in sera from heterologous or BNT vaccinated individuals.” So, if the booster doesn’t change antibody avidity, how could it induce sufficient antibody levels?

àAs indicated in Figure 4, the antibody levels strongly increased after the booster dose.

  1. Extensive English editing is necessary.

àA native speaker did the English editing and we removed some typos in the text.

  1. Overall, this is an interesting study. But here I found restrictions about some facts, like

Age restriction

The authors should justify the study for the above facts

àAlthough our cohort does not include younger (<27) or older (>65) participants, statements for middle aged individuals and therefore a significant fraction of the population can be made. We included the age limitation in the Discussion starting line 398 (s. above).

People suffering from other diseases

àWe agree that people suffering from other diseases are not covered by this study, but this was not the intention of our work. We analyzed healthy volunteers to test vaccine efficacy in vitro and obtain data useful to explain the high infection rates caused by Omicron. We included “with no relevant underlying health predispositions“ in line 75 to clarify this point.

Pregnant women

àNo pregnancies were reported during the course of the study. We included only healthy volunteers (s.above).

I trust that we addressed all the points raised by the reviewer. We thank you again for your review and hope that you find the revised manuscript now acceptable for publication.

Sincerely yours,

Prof. Dr. Barbara S. Schnierle

Reviewer 2 Report

The manuscript titled: “The SARS-CoV-2 variant Omicron is able to escape vaccine-induced humoral immune responses.” submitted to the journal Vaccines with manuscript ID: vaccines-1696105 by Hastert et al. have analyzed the specific binding of IgG to the RBD from WT, Delta or Omicron variants of SARS-CoV-2 in serum samples of vaccinated individuals.

The paper is interesting, fits the journal’s topic, has the potential to contribute to the knowledge about immune response against SARS-CoV variants, and is well written. However, there are issues that should be taken into account before publication.

My major concerns are with respect to methods, their presentation, and interpretation:

  • The sample size is limited. It should be overcome or at least discussed.
  • In the case of using the in-house ELISA, the authors should prove the specificity, sensitivity, cross-reactivity, and accuracy. Did the authors check the cross-reactivity with HCV, HBV, HIV, and CMV viruses? What are the advantages of this in-house ELISA over than commercially available ELISA test? There is no information about used positive or calibrator controls.
  • The cut-off level in ELISA and in serum neutralization assay was determined from the mean values obtained with samples taken prior to the J&J vaccination, which was from just 6 naïve individuals – one female and five male. However, the group of cases with a schedule of vaccination BNT/BNT/BNT are 5 females and 1 female. Another possible confounder is the age of the subjects. Did the authors check the possible age and sex differences? Are you referring to ethnicity?
  • Please provide the raw data of samples taken prior to the J&J vaccination with the deviations of the technical repeats, that are used for cut-off calculation.
  • Please, explain why the sera were pre-diluted at 1:50. Did you perform some serial dilutions priory?
  • Please, explain the calculation and the interpretation of the reciprocal area under the curve (1/AUC) value. There is no information about std. error; 95% Confidence Interval or others.
  • Please, clarify the calculation of the fold of RBD binding (line 301).
  • The following manuscript could be useful: Klein SL, Pekosz A, Park HS, Ursin RL, Shapiro JR, Benner SE, Littlefield K, Kumar S, Naik HM, Betenbaugh MJ, Shrestha R, Wu AA, Hughes RM, Burgess I, Caturegli P, Laeyendecker O, Quinn TC, Sullivan D, Shoham S, Redd AD, Bloch EM, Casadevall A, Tobian AA. Sex, age, and hospitalization drive antibody responses in a COVID-19 convalescent plasma donor population. J Clin Invest. 2020 Nov 2;130(11):6141-6150. doi: 10.1172/JCI142004.
  • The limitations of the study should be summarized.

Some minor points:

  • Each mean value should be presented with the corresponding standard deviation, including in the figures
  • Probably log-scale of the y-axis in Figures 2, 3, 4 would be better and could be easier to visualize

Author Response

Barbara Schnierle

Department of Virology

Section AIDS and newly emerging pathogens

Telefon / Phone +49 (0) 6103 77 5504

Fax: +49 (0) 6103 77 1234

E-Mail: [email protected]

May 11th, 2022

Paul-Ehrlich-Institut  Postfach  63207 Langen

Dear Reviewers,

First, we would like to thank the reviewers for the excellent comments and for giving us the opportunity to revise the manuscript. Below we listed the points raised by the reviewers, our response to them and the actions we took to advance the work.

Reviewer 2

The manuscript titled: “The SARS-CoV-2 variant Omicron is able to escape vaccine-induced humoral immune responses.” submitted to the journal Vaccines with manuscript ID: vaccines-1696105 by Hastert et al. have analyzed the specific binding of IgG to the RBD from WT, Delta or Omicron variants of SARS-CoV-2 in serum samples of vaccinated individuals.

The paper is interesting, fits the journal’s topic, has the potential to contribute to the knowledge about immune response against SARS-CoV variants, and is well written. However, there are issues that should be taken into account before publication.

My major concerns are with respect to methods, their presentation, and interpretation:

  • The sample size is limited. It should be overcome or at least discussed.

àAlthough our sample size is limited, it allows statements about middle-aged individuals (between 27-65 years of age) and therefor a major fraction of the populations. In addition, the obtained values for antibody binding or neutralization activity did not show large variations, which justifies our conclusions. We now included standard deviations in the figures and the text. In addition, we included a paragraph on limitations of the study in the Discussion starting at line 398:

“A limitation of the study is the small number of participants and potential effects from sex differences that cannot be accounted for in this study cohort. In addition, participants were healthy and between 27 and 65 year of age, which does not allow for conclusions on immune responses of individuals with comorbidities or children.”

  • In the case of using the in-house ELISA, the authors should prove the specificity, sensitivity, cross-reactivity, and accuracy. Did the authors check the cross-reactivity with HCV, HBV, HIV, and CMV viruses? What are the advantages of this in-house ELISA over than commercially available ELISA test? There is no information about used positive or calibrator controls.

à The used in-house ELISA was adapted from Amanat et al. 2020 [1]. According to this publication, no cross-reactivity against HIV and other coronaviruses (e.g. hCoV 229E and hCoV NL63) was detected. This ELISA is well established and was used in more than 100 publications over the last two years. We additionally added this publication in the Methods section. In addition, the study participants were healthy subjects with no relevant underlying health predispositions. We added this information in line 75.

The advantage of using this in-house ELISA is its rapid adaptation to the analysis of anti-RBD antibodies against different SARS-CoV-2 variants. Equal loading of the antigen onto the ELISA plate was performed with an antibody directed against the His-tag, which is included in RBD. We added this information in the manuscript in the Materials and Methods section starting at line 106.

  • The cut-off level in ELISA and in serum neutralization assay was determined from the mean values obtained with samples taken prior to the J&J vaccination, which was from just 6 naïve individuals – one female and five male. However, the group of cases with a schedule of vaccination BNT/BNT/BNT are 5 females and 1 female. Another possible confounder is the age of the subjects. Did the authors check the possible age and sex differences? Are you referring to ethnicity?
  •  

à Sex and age of all study participants are listened in Table 1. We agree with the reviewer that there are differences in sex distribution, especially between the adenoviral vaccinated and the heterologous or BNT vaccinated subjects. However, we mainly draw conclusions from data within one group. Most serological data of SARS-CoV-2 infections or vaccinations have a sex bias, since typically health care workers are analyzed who are to a high percentage females. We added this information and references in the Discussion starting in line 396: “We cannot totally exclude that these differences are due to sex differences in the different vaccine groups, because the adenovirus-based vaccine group comprised mainly male participants and the mRNA group mainly females. Interestingly, male convalescent COVID-19 patients had greater antibody titers than females and real world vaccine efficacy seems to be greater in men [2], [3].“

Data from Collier et al. (2021) [4], show that there is a significant difference in the humoral immune response of SARS-CoV-2 vaccinated subjects older than 80 years compared to younger subjects (20-79 years). Based on these findings, we expected no difference in humoral response in the age range of 20 to 79 years, which covers the age range analyzed here (27 to 65 years). All tested individuals are of Caucasian descent. This information was added in Table 1.

  • Please provide the raw data of samples taken prior to the J&J vaccination with the deviations of the technical repeats, that are used for cut-off calculation.

àThe raw OD450 values of the technical replicates with the respective standard deviations (SD) are summarized in the table below.

WT

Delta

Omicron

Replicate1

Replicate2

Replicate1

Replicate2

Replicate1

Replicate2

Sample1

0.06

0.059

0.053

0.053

0.051

0.051

Sample2

0.059

0.059

0.053

0.053

0.052

0.05

Sample3

0.08

0.079

0.08

0.078

0.078

0.08

Sample4

0.06

0.057

0.063

0.06

0.048

0.047

Sample5

0.056

0.056

0.065

0.062

0.047

0.045

The mean OD450 values and the resulting standard deviations were divided by the values obtained with the anti-His-tag antibody used to control equal loading of the RBD. The OD450 values were 2.45 for replicate 1 or 2.44 for replicate 2.

  • Please, explain why the sera were pre-diluted at 1:50. Did you perform some serial dilutions priory?
  •  

à Sera were diluted 1:50 to prevent overload of the ELISA binding capacity and to stay within the functional range of the assay. Different dilutions were tested and 1:50 resulted in the best results for different samples, including convalescent sera. We included this information in lines 106-107: “To prevent overload of the ELISA binding capacity and to stay within the functional range of the assay sera were pre-diluted 1:50 in PBS supplemented with 1% BSA and incubated for 2 h at RT.”

  • Please, explain the calculation and the interpretation of the reciprocal area under the curve (1/AUC) value. There is no information about std. error; 95% Confidence Interval or others.

à In brief, mean values from triplicate experiments and now the corresponding standard deviations were depicted in the graphs. For this 0 was chosen as baseline and only peaks above the baseline were considered positive. This information was added to the Material and Methods section (Statistical analysis and software) starting at line 147.

  • Please, clarify the calculation of the fold of RBD binding (line 301).

à Fold changes were calculated with by subtracting 1 from the quotient of the final value Y and the initial value X (Y/X-1). The information was added to Material and Methods (Statistical analysis and software) staring at line 154.

  • The following manuscript could be useful: Klein SL, Pekosz A, Park HS, Ursin RL, Shapiro JR, Benner SE, Littlefield K, Kumar S, Naik HM, Betenbaugh MJ, Shrestha R, Wu AA, Hughes RM, Burgess I, Caturegli P, Laeyendecker O, Quinn TC, Sullivan D, Shoham S, Redd AD, Bloch EM, Casadevall A, Tobian AA. Sex, age, and hospitalization drive antibody responses in a COVID-19 convalescent plasma donor population. J Clin Invest. 2020 Nov 2;130(11):6141-6150. doi: 10.1172/JCI142004.
  •  

àWe included the results on sex differences in SARS-CoV-2 directed immune responses in the discussion part. The following sentence was included: “We cannot totally exclude that these differences are due to sex differences in the different vaccine groups, because the adenovirus-based vaccine group comprised mainly male participants and the mRNA group mainly females. Interestingly, male convalescent COVID-19 patients had greater antibody titers than females and real world vaccine efficacy seems to be greater in men [2], [3].“

  • The limitations of the study should be summarized.

àWe concluded the study limitation in the discussion section and the following information was included starting in line 398: “A limitation of the study is the small number of participants and potential effects from sex differences that cannot be accounted for in this study cohort. In addition, participants were healthy and between 27 and 65 year of age, which does not allow for conclusions on immune responses of individuals with comorbidities or children.”

.”

  • Some minor points:

Each mean value should be presented with the corresponding standard deviation, including in the figures.

Standard deviations were added to all figures and in the text. New figures were introduced.

  • Probably log-scale of the y-axis in Figures 2, 3, 4 would be better and could be easier to visualize.

àIn our opinion, log-scale of the y-axis does not make the figures easier to visualize. For instance, the decline in SARS-CoV-2 specific antibodies after 6 month is better visible in our current version than in log-scale. Therefore, we did not change the y-axis in Figures 2, 3 and 4.

I trust that we addressed all the points raised by the reviewer. We thank you again for your review and hope that you find the revised manuscript now acceptable for publication.

Sincerely yours,

Prof. Dr. Barbara S. Schnierle

References

  1. Amanat F, Stadlbauer D, Strohmeier S, Nguyen THO, Chromikova V, et al. (2020) A serological assay to detect SARS-CoV-2 seroconversion in humans. Nature medicine 26 (7): 1033–1036. doi:10.1038/s41591-020-0913-5.
  2. Klein SL, Pekosz A, Park H-S, Ursin RL, Shapiro JR, et al. (2020) Sex, age, and hospitalization drive antibody responses in a COVID-19 convalescent plasma donor population. The Journal of clinical investigation 130 (11): 6141–6150. doi:10.1172/JCI142004.
  3. Bignucolo A, Scarabel L, Mezzalira S, Polesel J, Cecchin E, et al. (2021) Sex Disparities in Efficacy in COVID-19 Vaccines: A Systematic Review and Meta-Analysis. Vaccines 9 (8). doi:10.3390/vaccines9080825.
  4. Collier DA, Ferreira IATM, Kotagiri P, Datir RP, Lim EY, et al. (2021) Age-related immune response heterogeneity to SARS-CoV-2 vaccine BNT162b2. Nature 596 (7872): 417–422. doi:10.1038/s41586-021-03739-1.

Round 2

Reviewer 1 Report

The authors have addressed all the queries satisfactorily to their best level. A manuscript may now be accepted for publication in Vaccines.

Reviewer 2 Report

Thank you for your answers. 
The authors respond to my major concerns. 
Good luck with your future research.